# Role of the Hydroxyl Groups Coordinated toTiO$_2$ Surface on the Photocatalytic Decomposition of Ethylene at Different Ambient Conditions

Piotr Rychtowski [1], Beata Tryba [1,*], Agnieszka Skrzypska [1], Paula Felczak [1], Joanna Sreńscek-Nazzal [1], Rafał Jan Wróbel [1], Hiroyasu Nishiguchi [2] and Masahiro Toyoda [2]

1  Department of Catalytic and Sorbent Materials Engineering, Faculty of Chemical Technology and Engineering, West Pomeranian University of Technology in Szczecin, Piastów Ave. 42, 71-065 Szczecin, Poland; rp43903@zut.edu.pl (P.R.); agaskrzypska@op.pl (A.S.); pola.felczak@gmail.com (P.F.); joanna.srenscek@zut.edu.pl (J.S.-N.); rafal.wrobel@zut.edu.pl (R.J.W.)
2  Applied Chemistry, Faculty of Engineering, Oita University, 700 Dannoharu, Oita-Shi 870-1192, Japan; nishiguc@oita-u.ac.jp (H.N.); toyoda22@oita-u.ac.jp (M.T.)
*  Correspondence: beata.tryba@zut.edu.pl

**Abstract:** The titania pulp—a semi product received from the industrial production of titania white—was submitted for the thermal heating at 400–600 °C under Ar and H$_2$ to obtain TiO$_2$ with different structure and oxygen surface defects. Heating of titania in H$_2$ atmosphere accelerated dehydration and crystallisation of TiO$_2$ compared to heating in Ar. TiO$_2$ prepared at 500 and 600 °C under H$_2$ had some oxygen vacancies and Ti$^{3+}$ centres (electron traps), whereas TiO$_2$ obtained at 450 °C under H$_2$ exhibited some hole traps centres. The presence of oxygen vacancies induced adsorption of atmospheric CO$_2$. It was evidenced, that ethylene reacted with TiO$_2$ after UV irradiation. Formic acid was identified on TiO$_2$ surface as the reaction product of ethylene oxidation. Hydroxyl radicals were involved in complete mineralisation of ethylene. TiO$_2$ prepared at 500 °C under H$_2$ was poorly active because some active sites for coordination of ethylene molecules were occupied by CO$_2$. The most active samples were TiO$_2$ with high quantity of OH terminal groups. At 50 °C, the physically adsorbed water molecules on titania surface were desorbed, and then photocatalytic decomposition of ethylene was more efficient. TiO$_2$ with high quantity of chemisorbed OH groups was very active for ethylene decomposition. The acidic surface of TiO$_2$ enhances its hydroxylation. Therefore, it is stated that TiO$_2$ having acidic active sites can be an excellent photocatalyst for ethylene decomposition under UV light.

**Keywords:** ethylene decomposition; reduced TiO$_2$; titania surface defects

## 1. Introduction

Ethylene has a wide range of applications in the world, being used as an anesthetic, a welding gas, and a substrate for making polyethylene, ethylene oxide, or styrene. Ripening fruits such as bananas and apples emit ethylene. Ethylene accumulation during fruit storage can result in premature ripening, bitter taste, loss of chlorophyll, or increased susceptibility to disease [1,2]. Effective ethylene removal in such areas is, therefore, extremely important. These situations can be prevented by using materials or coatings that break down ethylene into simple compounds. One of the methods is use of coatings containing titanium dioxide, which in stages decomposes ethylene to carbon dioxide and water [3–8].

Photocatalytic decomposition of ethylene depends not only on the structure of the titania material but also the ambient conditions, such as humidity, temperature and concentration of oxygen [2–5,9]. It was reported that in the mechanism of ethylene decomposition, the photo-formed OH radicals as well as O$_2^-$ and O$_3^-$ anion radicals were the crucial species responsible for its complete photocatalytic oxidation into CO$_2$ and H$_2$O [4]. Hydroxyl radicals could be generated at the presence of water vapor or hydroxyl ions adsorbed

on the titania surface; however, in the excess of humidity, the conversion of ethylene is decreased [2,4,9]. The most probable at high concentration of water molecules in the air, under UV irradiation, the titania surface is completely covered with water due to the superhydrophilic effect, and then uptake of oxygen on active titania sites is limited [4]. Adsorption of oxygen on $TiO_2$ is necessary for generation of superoxide anionic radicals and suppression of charge carriers' recombination. Some of the researchers observed a strong increase in the reaction rate between 30 and 65 °C, attributed to a significant decrease in the adsorption of water molecules which compete with ethylene adsorption [2]. Maximum photocatalytic ethylene oxidation rates were obtained between 100 and 200 °C, and this was related to the differences in the ethylene and water adsorption energies on the polar Ti-OH surface. Decrease in the photocatalytic activity was observed above 200 °C due to the loss of the photogenerated charge carriers, resulting from the nonradiative, multiphonon recombination [2]. During photocatalytic decomposition of ethylene water, molecules are formed and can contribute to adsorption on the titania surface [9]. It was proved that when the photocatalytic decomposition of ethylene was carried out at 32 °C, then the rate of ethylene conversion gradually decreased from the initial one during 3 h up to reaching the steady state. After 3 h, water molecules started to be present in the reaction chamber. The initial decline in ethylene conversion was caused by adsorption of water molecules on the titania surface [9]. Quite a different situation took place when the photocatalytic decomposition was carried out at 108 °C. Then all the water molecules originated from ethylene decomposition were not adsorbed on the titania surface, and no drop in the ethylene decomposition upon the time of UV irradiation was observed [9].

It was reported that the excessive oxygen is necessary for complete decomposition of ethylene [4]. It was suggested that initially ethylene is oxidized to CO and then to $CO_2$ by the excessive oxygen [4]. The reaction mechanism can be expressed as follows:

$$O^* + C_2H_4 \rightarrow (C_2H_4O)^* \rightarrow CO \rightarrow CO_2 \tag{1}$$

From the other hands, Yamazaki et al. [10] proposed the other mechanism, in which adsorbed ethylene reacts with OH radicals to form $C_2H_4OH^\bullet$ intermediate radicals that can subsequently react with adsorbed oxygen to get final mineralization into $CO_2$, according to the following reactions:

$$C_2H_4 + \sigma \rightarrow C_2H_4\sigma \tag{2}$$

$$H_2O + \sigma \rightarrow H_2O\sigma \tag{3}$$

$$O_2 + \sigma' \rightarrow O_2\sigma' \tag{4}$$

$$H_2O\sigma + h^+{}_{VB} \rightarrow OH\sigma + H^+\sigma \tag{5}$$

$$OH\sigma + C_2H_4\sigma \leftrightarrow C_2H_4OH\sigma \tag{6}$$

$$C_2H_4OH\sigma + O_2\sigma' \rightarrow \text{mineralization to } CO_2 \tag{7}$$

where $\sigma$ and $\sigma'$ indicate different types of active sites at the surface of $TiO_2$.

Oxygen and water molecules occupy different active sites on the titania surface, and both of them are sources for generation of reactive radicals, which are utilised for the photocatalytic decomposition of ethylene.

The other mechanism of ethylene decomposition over $TiO_2$ and UV irradiation was introduced by Hauchecorne et al. [11], who did FTIR measurements of titania surface in situ during photocatalytic decomposition of ethylene. They suggested that ethylene was going decomposition through the formation of two intermediates—formaldehyde and formic acid—for which formaldehyde was bound in two different ways (coordinatively and as bidentate). Finally, $CO_2$ and $H_2O$ were found as end products, resulting in the complete mineralisation of the pollutant [11]. They summarised that hydroxyl radicals were largely used for total mineralisation of ethylene, as it was illustrated in the stoichiometric reaction below:

$$C_2H_4 + 12OH^\bullet \rightarrow 2CO_2 + 8H_2O \tag{8}$$

Hydroxyl radicals were formed upon reaction of terminated hydroxyl groups on titania surface with the photoinduced holes. During photocatalytic decomposition of ethylene, decreasing of surface hydroxyl groups on $TiO_2$ was observed [11]. The other researchers pointed out the key role of the adsorbed water on $TiO_2$ surface in the ethylene decomposition [3]. They observed that after complete drying of the titania, the ethylene degradation was significantly reduced.

It was largely reported that anatase type $TiO_2$ with a large surface area, large band gap, and numerous OH groups was efficient for the oxidation of ethylene [3,4]. The other researchers indicated that high photocatalytic decomposition of ethylene could be achieved on $TiO_2$ having high quantity of surface defects, which were formed upon calcination of $TiO_2$ at low temperature, such as 450 °C in the excess of oxygen [12]. They proved formation of $Ti^{3+}$ sites, which were responsible for the effective separation of free charges due to the trapping of the photoinduced electrons and increased adsorption of the oxygen molecules on titania surface [12]. The other researchers investigated the role of the coordination of titanium (Ti) atom on the surface [13]. $TiO_2$ with exposed (001) face exhibits 50% of five-coordinated Ti ($Ti_{5c}$) atoms, while that with exposed (101) surface exhibits 100% of $Ti_{5c}$ atoms. Thus, the titania structure with (001) facets was considered to be more reactive for the surface catalyzed reactions. However, performed research studies indicated poor photocatalytic activity of the prepared titania with (001) facets towards ethylene decomposition, contrary to acetaldehyde [13]. Ethylene demonstrated lower adsorption energy than acetaldehyde on the $TiO_2$ (001) and indicated the weak interaction with titania surface [13]. Therefore, there is a challenge to design photocatalytic material for indoor air purification with excellence properties towards all the VOCs commonly present in the environment. In our previous paper [14], we have presented properties of the reduced $TiO_2$ towards acetaldehyde decomposition under irradiation of the fluorescent lamp. In this paper, hydrogen reduced $TiO_2$ was tested for ethylene decomposition under fluorescent lamp irradiation and the various UV lamps irradiations at the reactor chamber temperatures of 25 and 50 °C. Impact of the titania parameters such as hydroxylation of surface, presence of defects, anatase crystallites size and specific surface area on its photocatalytic activity towards ethylene decomposition will be presented and discussed.

## 2. Results

The physicochemical characteristics of the studied samples were introduced in Table 1. In Figure 1, XRD patterns are shown.

**Table 1.** Brief characteristics of the titania samples heat-treated under $H_2$ and Ar.

| Sample Name | HTT (°C) | BET Surface Area (m$^2$/g) | Phase Composition (Anatase:Rutile) | Average Crystallites Size of Anatase (nm) |
|---|---|---|---|---|
| TiO$_2$_A150 | - | 215 | 95:5 | 15.2 |
| TiO$_2$_400-H$_2$ | 400 | 155 | 97:3 | 15.1 |
| TiO$_2$_450-H$_2$ | 450 | 130 | 96:4 | 16.3 |
| TiO$_2$_500-H$_2$ | 500 | 81 | 96:4 | 24.3 |
| TiO$_2$_600-H$_2$ | 600 | 40 | 92:8 | 35.2 |
| TiO$_2$_400-Ar | 400 | 167 | 97:3 | 15.0 |
| TiO$_2$_450-Ar | 450 | 139 | 95:5 | 15.5 |
| TiO$_2$_500-Ar | 500 | 123 | 96:4 | 16.8 |
| TiO$_2$_600-Ar | 600 | 68 | 96:4 | 25.5 |

Sample denoted as $TiO_2$_A150 was obtained during hydrothermal treatment of titania pulp in autoclave at 150 °C for 1 h; preparation details were published elsewhere [15]. The other samples were obtained through the thermal heating of $TiO_2$_A150 under $H_2$ or Ar atmospheres. Heating of $TiO_2$ under hydrogen accelerated both crystallisation of anatase and transformation of anatase into rutile by comparison with heat treatment in Ar. High increase in anatase crystallites was observed after heating titania at 500 °C under $H_2$ and

at 600 °C in Ar. BET surface area of titania samples was greatly diminished when rapid growth of anatase crystallites took place.

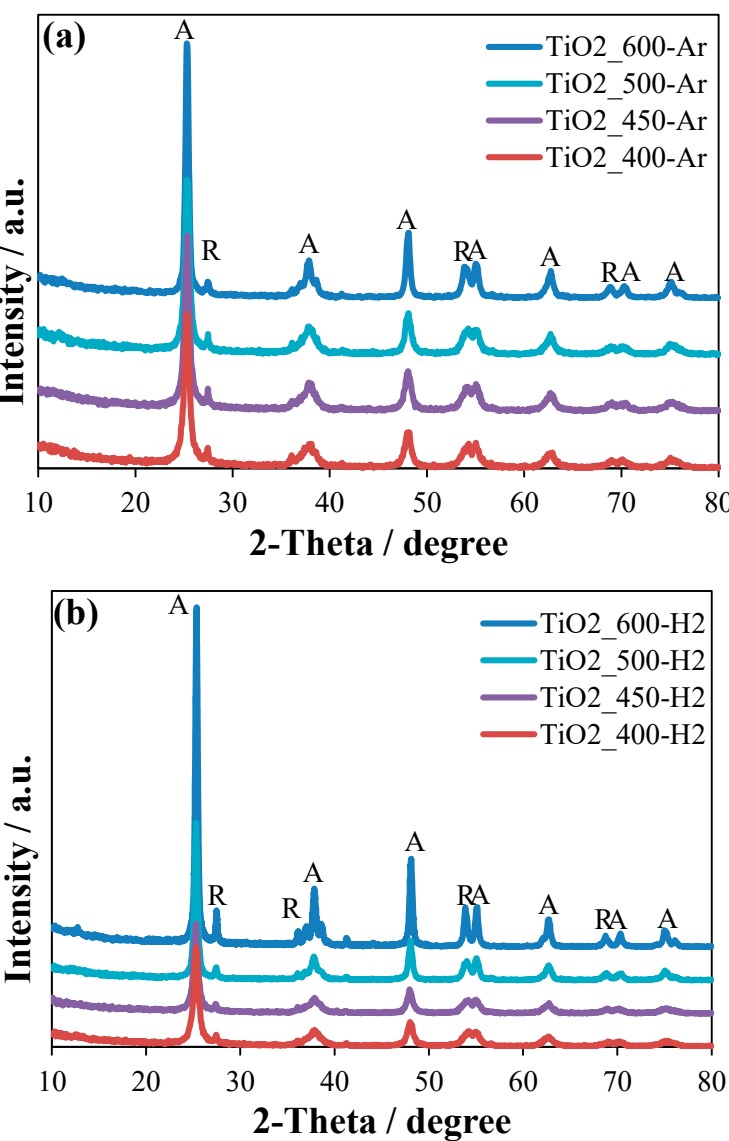

**Figure 1.** XRD patterns of the prepared TiO$_2$ samples heating at 400–600 °C under (**a**) Ar, (**b**) H$_2$.

Hydrogen treatment of TiO$_2$ was performed in order to introduce some surface defects. In the previous published paper, we have presented some EPR spectra for titania samples heat-treated at 450–600 °C in Ar and at 400–500 °C under hydrogen [14]. Some spins related to the O$^{\bullet-}$ radicals were observed in titania samples reduced at 450–500 °C in H$_2$ and those treated at 450 and 600 °C in Ar. Sample heat treated at 500 °C in H$_2$ showed some electron traps centres. However, the observed signal in EPR spectrum was very low intensity [14]; therefore, some new measurements were performed for the samples reduced in H$_2$ at 500 and 600 °C using high sensitive apparatus. The EPR spectra obtained for these samples were shown in Figure 2.

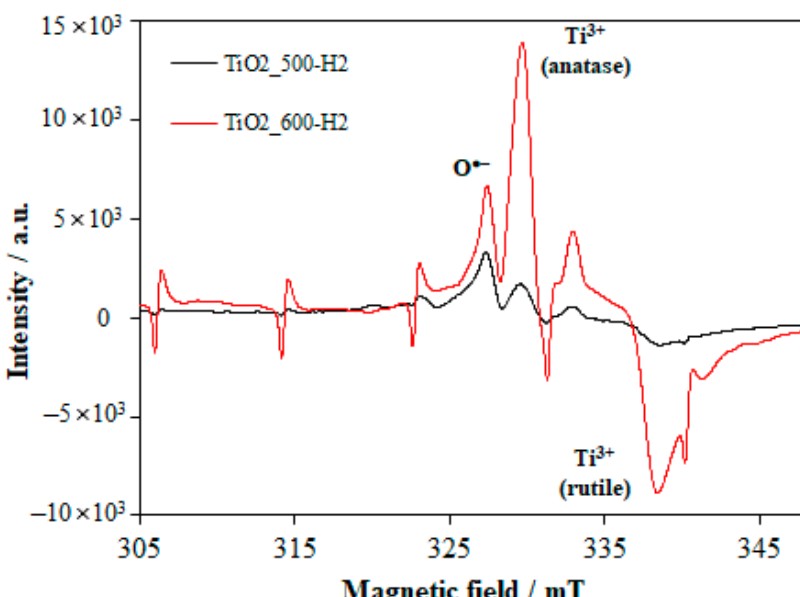

**Figure 2.** EPR spectra of TiO$_2$ samples heat treated at 500 and 600 °C in H$_2$.

The signal at g = 2.002 can be assigned to the Ti$^{4+}$O$^{2-}$Ti$^{4+}$O$^{\bullet-}$ radicals in anatase, and that at g = 1.988 to electron traps (Ti$^{3+}$) in anatase and at g = 1.94 − Ti$^{3+}$ in rutile [16]. These measurements showed clearly that some Ti$^{3+}$ centres were formed in TiO$_2$ heated at 500 °C in H$_2$, and they are of higher intensity in TiO$_2$ sample heated at 600 °C. Although TiO$_2$ obtained at 600 °C in H$_2$ contained only 8% of rutile, the signal of surface Ti$^{3+}$ in rutile was of high intensity, most likely due to the higher reduction abilities of rutile than anatase [17] and higher stability of Ti$^{3+}$ in rutile by comparison with an anatase [12].

The presence of the hydroxyl groups on the titania surface was analysed by FTIR Spectroscopy; in Figure 3, FTIR spectra recorded for the titania samples are presented.

The broad absorption band in the range of 3600–2500 cm$^{-1}$ can be assigned to the ν(OH) of water molecules and hydroxyls interacting via hydrogen bond, while the signal at around 3700 cm$^{-1}$ to free OH groups of H$_2$O molecules pointing out from the water surface multilayer [18]. Intensity of this broad band decreases with an increase in temperature of heat treatment, due to the surface dehydration. The band with the maximum at around 1630 cm$^{-1}$ is assigned to the water bending mode [18]. The intensity of the band at 1630 cm$^{-1}$ seems to be comparable among the samples prepared in Ar; however, somewhat lower intensity can be noticed for sample heat treated at 500 °C. In case of hydrogenated titania samples, that one prepared at 450 °C seems to have the highest intensity band at 1630 cm$^{-1}$, but those prepared at 500 and 600 °C exhibited additional a small intensity band at 1552 cm$^{-1}$, which can be assigned to some -COO groups assigned to CO$_2$ adsorbed on the defect sites [14].

FTIR analyses did not allow to quantify the amount of hydroxyl groups bounded with titania surface through mono and multilayer. XPS measurements were performed to analyse both the presence of the titania surface defects and amount of the surface hydroxyl groups on the titania surface In Figure 4, there are presented some XPS spectra for titania samples prepared under hydrogen treatment.

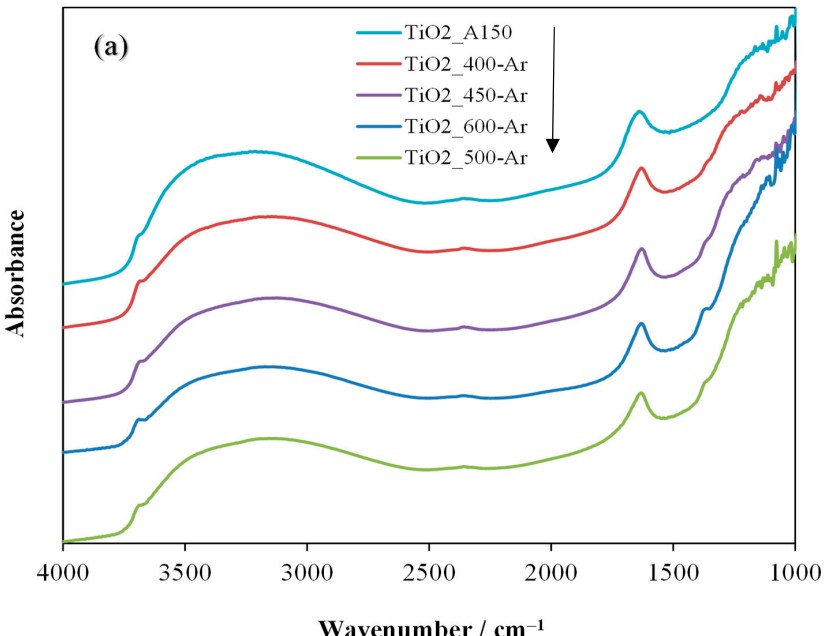

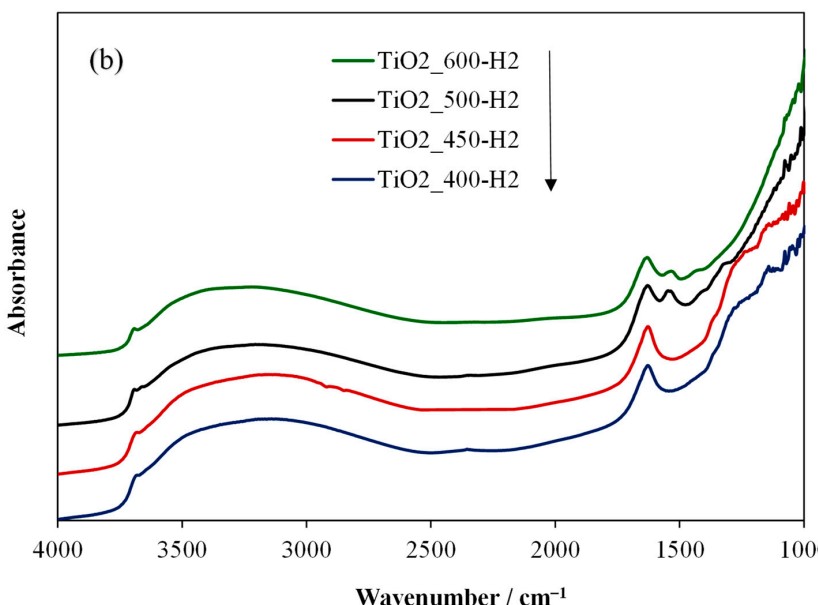

**Figure 3.** FTIR spectra for TiO$_2$ heat treated at 400–600 °C under (**a**) Ar, (**b**) H$_2$.

Ti 2p signal consists of two peaks, Ti2p 1/2 and Ti2p 3/2, with binding energy of 464.4 and 458.6 eV, respectively. The peak at 458.6 eV is slightly asymmetric due to the presence of Ti$^{3+}$ species, observed after titania reduction. The calculated quantities of Ti$^{3+}$ on the surface of samples prepared at 450, 500 and 600 °C were equal at 2.7, 4.5, and 3.7%, respectively. Although EPR measurements showed high quantity of Ti$^{3+}$ centres in titania hydrogenated at 600 °C in the bulk, the sample prepared at 500 °C exhibited the highest quantity of Ti$^{3+}$ on the surface. A significant difference in the surface hydroxyl groups can be noticed between titania hydrogenated at 450 °C and those obtained at higher temperatures, such as 500 and 600 °C. Calculated concentrations of the surface OH groups on the hydrogenated titania samples were equal at 17.2, 4.6 and 6.2%, for TiO$_2$ prepared at 450, 500 and 600 °C, respectively. All the titania samples heat-treated in Ar showed relatively high concentration of the surface OH groups, from 17.5% at. for sample heated at 600 °C to 20.5% at. for that prepared at 450 °C.

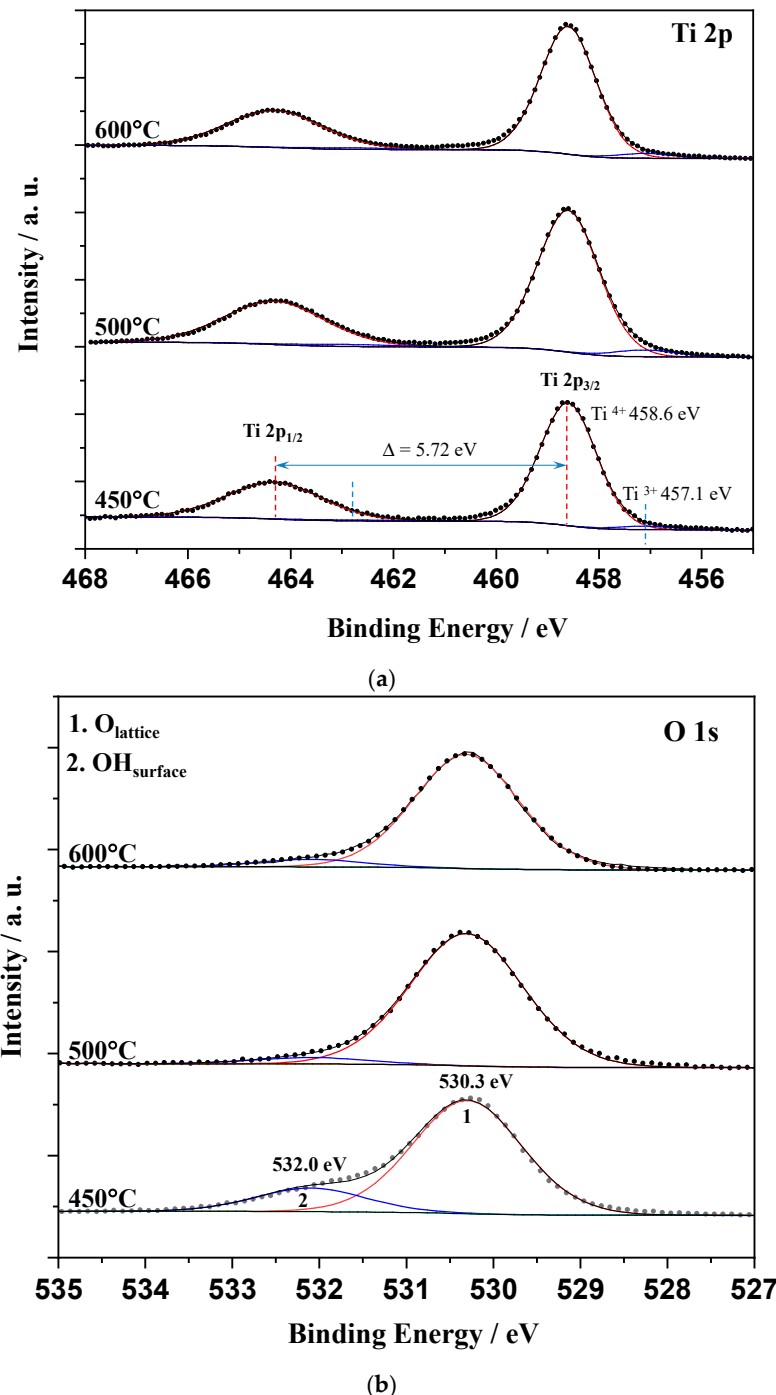

**Figure 4.** XPS spectra of TiO$_2$ heat treated at 450–600 °C under hydrogen, (**a**) Ti2p, (**b**) O1s.

Detailed determination of the OH groups on TiO$_2$ surface was performed by TG analyses. By using the proper temperature program, it was possible to differentiate OH groups, which were weak and strong bounded with titania surface. The impact of the bridging and terminal OH groups on the photocatalytic activity of TiO$_2$ was described elsewhere [19]. In Table 2, the percentage weights of physisorbed and chemisorbed OH groups on TiO$_2$ surface were listed.

In general, samples prepared under H$_2$ treatment contained less OH groups than those heat-treated in Ar at the same temperature. With an increase in temperature of heating, the quantity of OH groups on TiO$_2$ surface decreased.

**Table 2.** Amount of OH groups on $TiO_2$ surface determined by TG analyses.

| Sample | OH Physisorbed (wt%) | OH Chemisorbed (wt%) | Total Mass Loss (wt%) |
|---|---|---|---|
| $TiO_2$_400-Ar | 2.28 | 1.84 | 4.13 |
| $TiO_2$_450-Ar | 1.30 | 0.91 | 2.21 |
| $TiO_2$_500-Ar | 1.17 | 0.96 | 2.13 |
| $TiO_2$_600-Ar | 0.93 | 0.38 | 1.31 |
| $TiO_2$_400-$H_2$ | 1.99 | 1.60 | 3.59 |
| $TiO_2$_450-$H_2$ | 1.17 | 0.51 | 1.68 |
| $TiO_2$_500-$H_2$ | 0.96 | 0.45 | 1.41 |
| $TiO_2$_600-$H_2$ | 0.55 | 0.16 | 0.71 |

Higher adsorption of hydroxyl groups on the titania surface can be supported by the acidity of the titania surface [20]. The raw material used for studies was obtained from the industrial production of titania white in Poland, which is running based on the sulphuric method. The raw titania in a form of a white pulp contained a low percentage of sulphates. The percentage of sulphur in the studied titania samples was determined by XRFS technique. The results from the measurements were summarised in Table 3.

**Table 3.** Sulphur content in titania samples measured by XRFS.

| Sample | Content of Sulphur |
|---|---|
| Titania pulp | 2.1 wt% |
| $TiO_2$_A150 | 1.5 wt% |
| $TiO_2$_400-Ar | 1.40 wt% |
| $TiO_2$_450-Ar | 1.46 wt% |
| $TiO_2$_500-Ar | 1.39 wt% |
| $TiO_2$_600-Ar | 0.76 wt% |
| $TiO_2$_400-$H_2$ | 1.45 wt% |
| $TiO_2$_450-$H_2$ | 0.23 wt% |
| $TiO_2$_500-$H_2$ | 0.20 wt% |
| $TiO_2$_600-$H_2$ | 821 ppm |

Heat treatment of $TiO_2$ under hydrogen gas caused faster removal of sulphates from the titania raw material. High drop in sulphur content was observed in $TiO_2$ samples heated in $H_2$ above 400 °C, which revealed high quantity of surface defects.

In the next step, the acidity of the titania surface was analysed through the performance of acid base titration curve for pair of titania samples heat treated at 450 °C under $H_2$ and Ar. The results are presented in Figure 5.

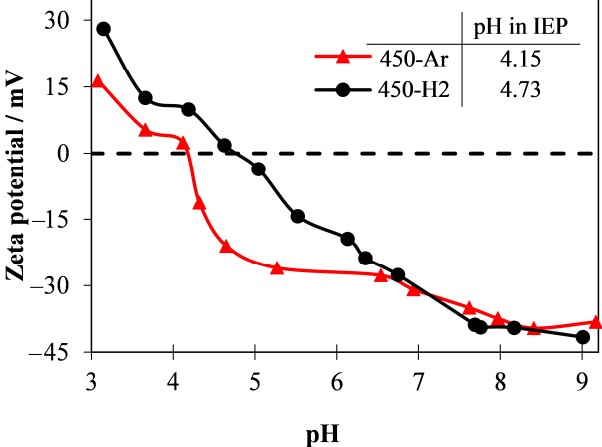

**Figure 5.** The zeta potential versus pH of $TiO_2$_450-Ar and $TiO_2$_450-$H_2$ suspensions.

Sample prepared in Ar showed shifting of IEP towards lower pH. It means that this sample exhibited more acidic sites on the surface than that one heat treated in $H_2$. Higher acidic surface of titania prepared in Ar could be caused by the presence of sulphates, which were of higher amounts than in hydrogenated titania sample. It was reported that the presence of anionic species on titania surface enhanced formation of free hydroxyl radicals [20]. Moreover, these anionic species can force the movement of photogenerated holes to the titania surface due to the electrostatic attraction, so they can improve separation of charge carriers and facilitate formation of hydroxyl radicals.

Hydrogenation of titania can conduct to changes in its optical properties through the formation of some localized electron states induced by formation of oxygen vacancies and $Ti^{3+}$ defects [21]. The colour of the titania hydrogenated samples can change from white to pale yellow, yellow, brown, blue, grey or black, depending on the preparation methods and conditions [21]. The prepared titania samples under hydrogen treatment above 450 °C revealed change of colour onto brownish (at 500 °C) and grey (at 600 °C). The UV-Vis/DR absorption spectra of the studied samples were illustrated in Figure 6.

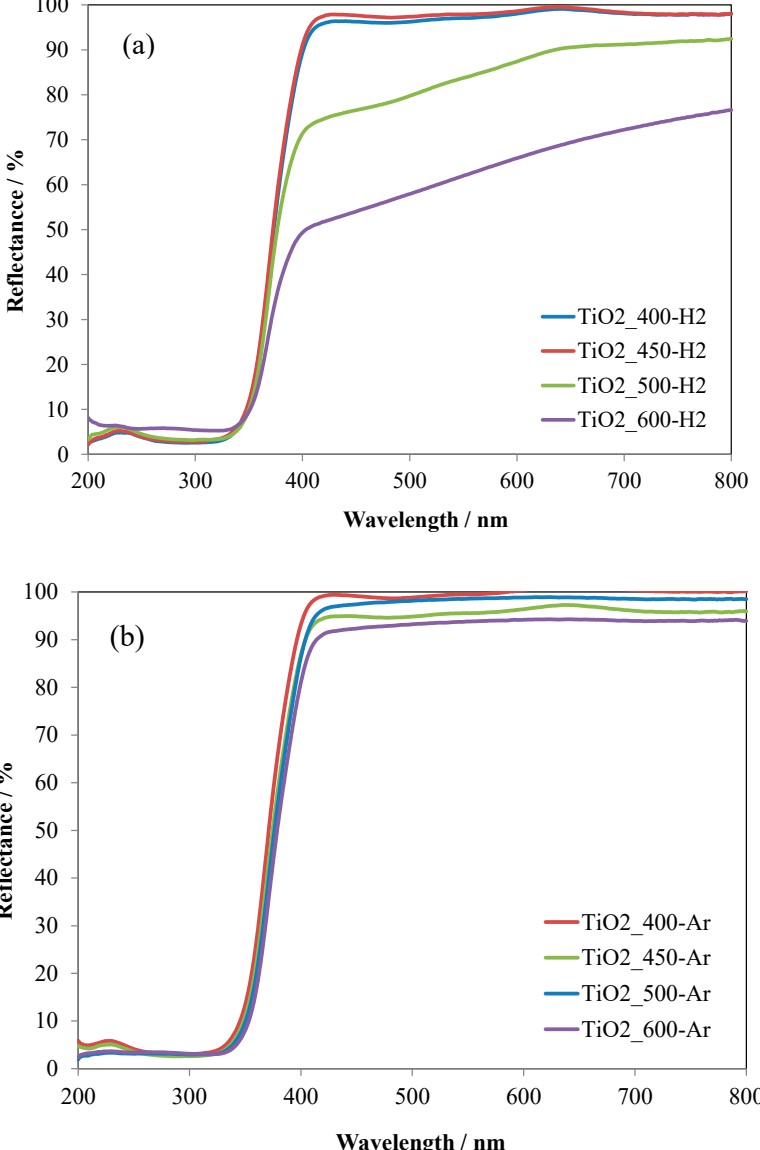

**Figure 6.** UV-Vis/DR spectra of titania samples heat-treated at 400–600 °C under (**a**) $H_2$, (**b**) Ar.

The performed measurements indicated that the coloured samples exhibited absorption of the visible light, which was higher for the sample prepared at 600 °C, having the highest number of $Ti^{3+}$ structural defects. This sample also showed somewhat shifting of the absorption edge towards visible light, which was probably related to the increase in the rutile content and formation of some electron density states below the conduction band.

The photoluminescence spectra were measured for the studied titania samples in order to check the recombination process occurring between electron and hole pairs after excitation with UV light.

The photoluminescence spectra are presented in Figure 7. Titania sample $TiO_2$_A150, which was obtained in the hydrothermal process at 150 °C without further thermal treatment, showed the highest intensity peak of the photoluminescence emission. This sample contained higher quantity of physisorbed water than the others. It was already proved that, formed upon excitation of $TiO_2$, charge carriers on high hydroxylated anatase surface with multilayers of water molecules could follow recombination process faster than on the dry one [22]. The sample prepared at 450 °C in Ar showed quite low intensity emission peak; at this temperature, complete dehydration of physisorbed water usually occurs [23]. Samples, which revealed defected structure in the form of hole traps, showed high photoluminescence peak. Highly reduced $TiO_2$ sample obtained at 600 °C in $H_2$ also showed high photoluminescence. However, the photoluminescence emission was very low for $TiO_2$ heat treated at 500 °C under $H_2$, most likely coexisting both kinds of defects such as hole and electron traps that could induce enhance separation of free radicals.

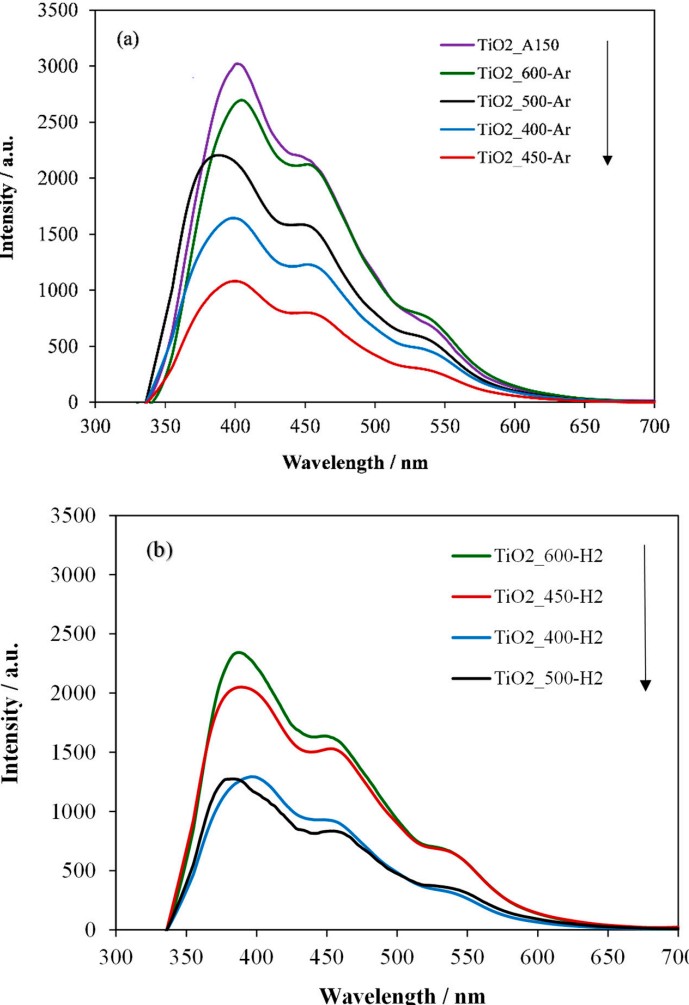

**Figure 7.** Photoluminescence spectra measured after excitation of titania at λ = 390 nm, (**a**) $TiO_2$ after hydrothermal treatment at 150 °C and these heat treated in Ar and (**b**) $TiO_2$ heat treated in $H_2$.

The morphology of the prepared TiO$_2$ samples was analysed by SEM technique. In Figure 8, some of the SEM images of TiO$_2$ samples were illustrated.

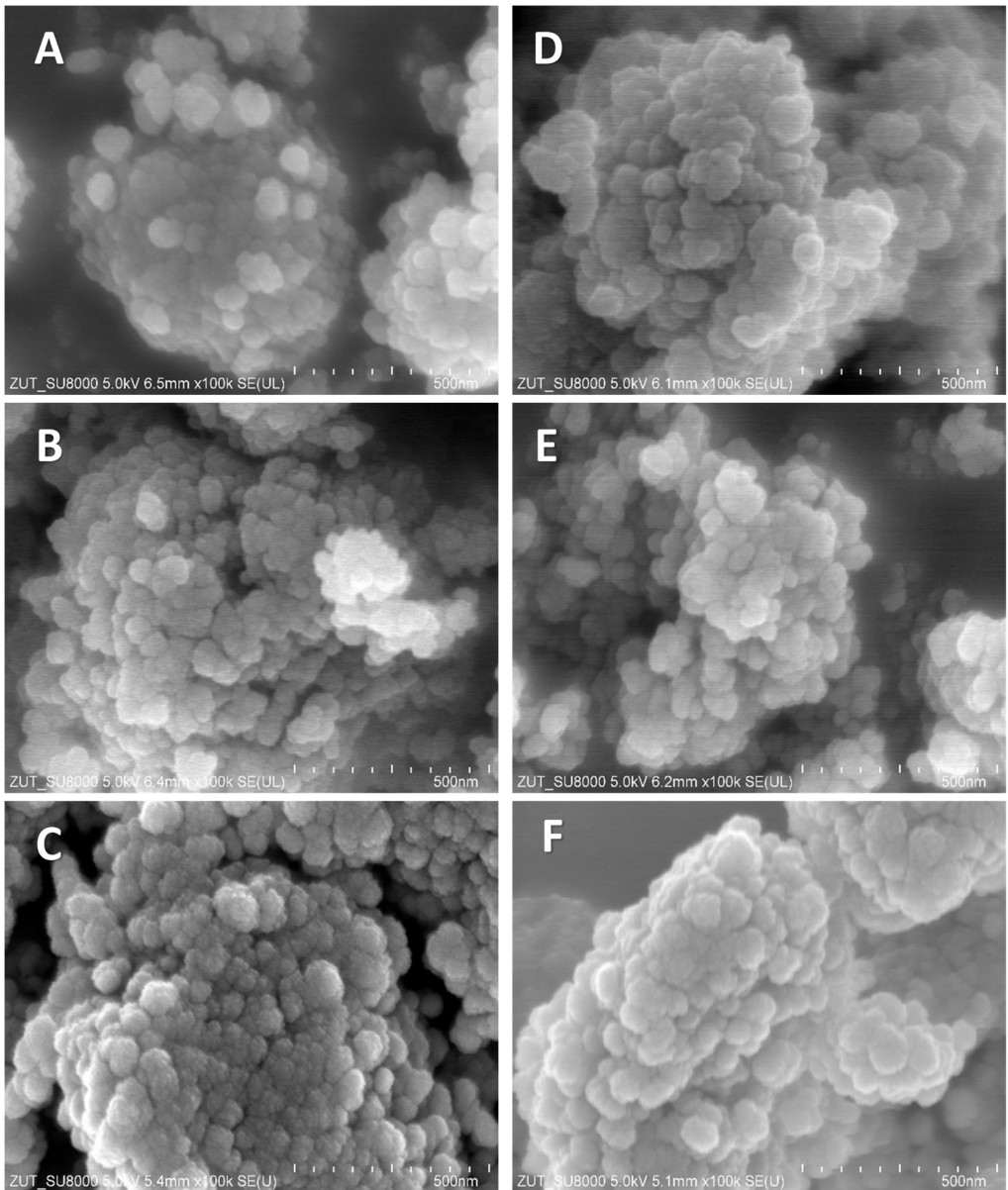

**Figure 8.** SEM images of TiO$_2$ prepared in Ar at (**A**) 400 °C, (**B**) 500 °C, (**C**) 600 °C and at H$_2$ (**D**) 450 °C, (**E**) 500 °C, (**F**) 600 °C, magnification ×100 k.

The structure of all the TiO$_2$ samples was quite comparable; some agglomerates consisted from small particles are visible. Sample prepared at 600 °C in H$_2$ showed somewhat higher size agglomerates than the others. SEM images of higher magnification were placed as Supporting Information Figure S1.

All the prepared samples were tested for the photocatalytic decomposition of ethylene at different conditions. Detailed description of the installation set up can be found in our previous paper [14]. For the photocatalytic test performed at temperature 50 °C, the other quartz tube reactor and different UV source were used, and some luminous UV tubes were applied. In Table 4, there are introduced the results of ethylene decomposition after 1 h of irradiation on the prepared titania samples at different conditions.

**Table 4.** Percentage decomposition of ethylene on the titania samples after 1 h of irradiation under different conditions.

| Sample Name | Conditions of the Photocatalytic Test | | |
|---|---|---|---|
| | $C_2H_4$ Concentration 10 ppm Temperature 25 °C Lamp-Fluorescent Flow Rate: 20 mL/min | $C_2H_4$ Concentration 50 ppm Temperature 25 °C Lamp-UV-1 Flow Rate: 50 mL/min | $C_2H_4$ Concentration 50 ppm Temperature 50 °C Lamp-UV-2 Flow Rate: 50 mL/min |
| | Ethylene Decomposition after 1 h of Irradiation (%) | | |
| $TiO_2$_A150 | 30 | - | 66 |
| $TiO_2$_400-Ar | 93 | 94 | 87 |
| $TiO_2$_450-Ar | 86 | 92 | 86 |
| $TiO_2$_500-Ar | 31 | 96 | 85 |
| $TiO_2$_600-Ar | 83 | 92 | 80 |
| $TiO_2$_400-$H_2$ | 30 | 92 | 90 |
| $TiO_2$_450-$H_2$ | 30 | 86 | 92 |
| $TiO_2$_500-$H_2$ | 3 | 53 | 40 |
| $TiO_2$_600-$H_2$ | 9 | 81 | 62 |

Titania samples heat treated in Ar at low temperatures, such as 400–450 °C, revealed high photocatalytic decomposition of ethylene under irradiations of both, fluorescent and UV lamps. Contrary to that, samples prepared under hydrogen treatment showed poor activity towards ethylene decomposition when a fluorescent lamp was used, but their activity was greatly enhanced under UV irradiation at the reaction temperature of 50 °C. However, highly reduced samples, prepared at 500 and 600 °C under hydrogen exhibited low photocatalytic activity, even under UV irradiation. Titania sample obtained after hydrothermal treatment in autoclave at 150 °C showed low ethylene decomposition under irradiation of a fluorescent lamp; however, its activity was increased double under UV irradiation at temperature of 50 °C, nevertheless, was still lower in comparison with the other titania samples. This is very interesting that sample $TiO_2$_450-$H_2$ showed enhanced photocatalytic activity under UV irradiation at 50 °C.

It was reported that in the absence of light irradiation, $C_2H_4$ did not adsorb nor dissociate with $TiO_2$ [24]. Weak interaction of $C_2H_4$ with commercial $TiO_2$-P25 was noted; however, some studies indicated adsorption of ethylene on a reduced anatase $TiO_2$ (001) surface via oxygen vacancy sites [24].

In order to check the interaction of ethylene with titania surface in situ, diffuse reflectance infrared Fourier transform spectroscopy (DRIFTS) was applied with using Praying Mantis DRIFTS accessory. Detailed description of this accessory was reported elsewhere [25]. For the purpose of these measurements, higher concentration of ethylene gas in air was used (200 ppm). The sample surface was irradiated through the quartz window of reactor by the UV LED light-emitting diode having an optical power of 415 mW. Simultaneously FTIR spectra of the sample surface were recorded within the time of the running process. In Figure 9, some of the selected spectra from the adsorption and photocatalytic decomposition of ethylene on two samples: $TiO_2$_400-Ar and $TiO_2$_500-$H_2$ were shown. Recognitions of the IR bands were performed on the basis of the IR spectra library.

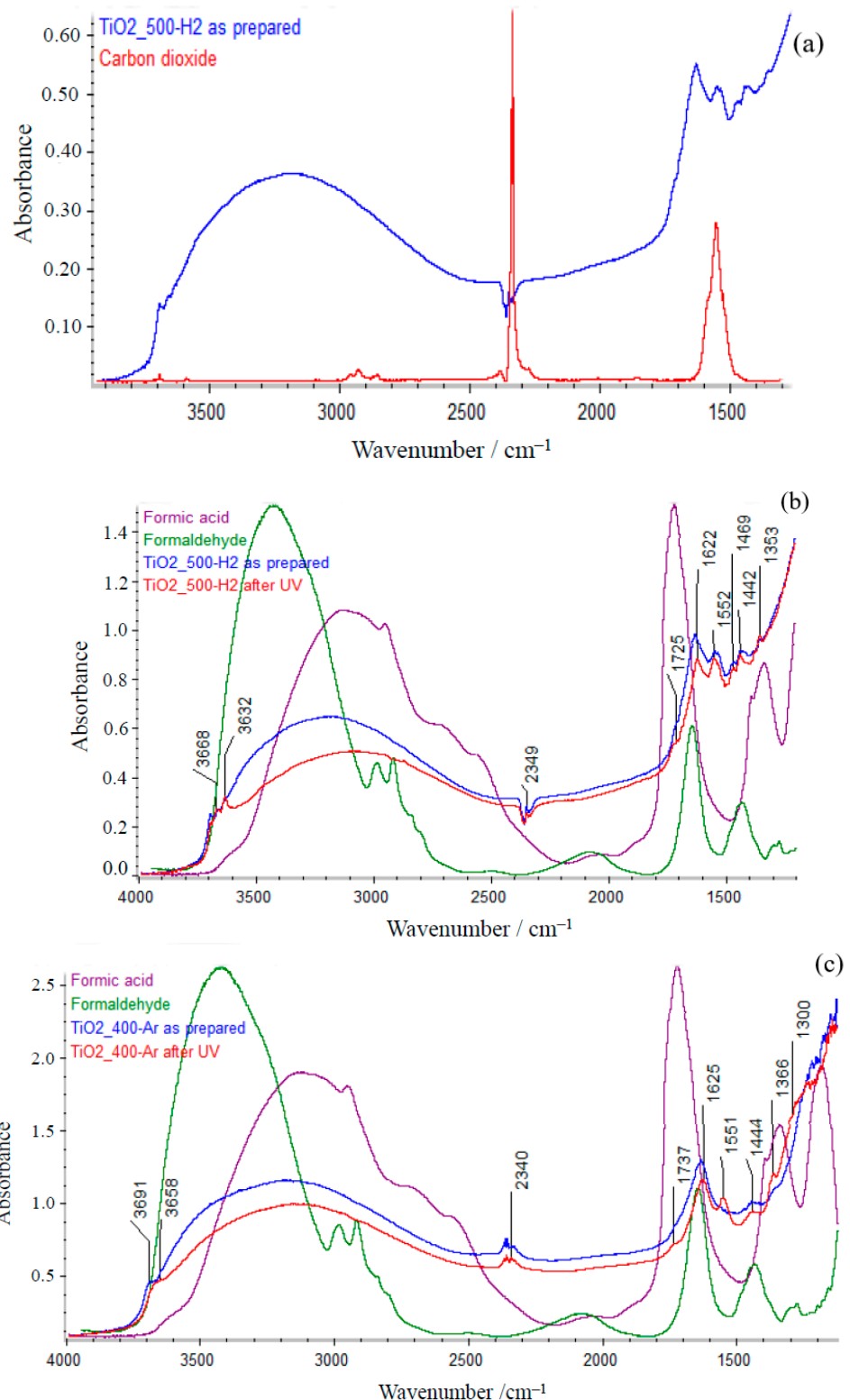

**Figure 9.** FTIR spectra from the in situ measurements of ethylene decomposition under UV irradiation with comparison of selected IR spectra patterns of iced $CO_2$, formic acid and formaldehyde, (**a**) $TiO_2\_500$-$H_2$ as prepared, (**b**) $TiO_2\_500$-$H_2$ before and after UV irradiation and (**c**) $TiO_2\_Ar$-400 before and after UV irradiation.

The chemical composition of $TiO_2\_500$-$H_2$ surface after preparation contained not only the bands related to OH groups (at 3700–2500 and at 1630 cm$^{-1}$) but also the band at

$1552$ cm$^{-1}$, which was assigned to $\nu_{as}$COO vibrations in $CO_2$ adsorbed on $TiO_2$ surface, Figure 9a. The most likely $CO_2$ was adsorbed at the oxygen vacancies sites. This band was not detected in $TiO_2$ hydrogenated under low temperature such as 400–450 °C (Figure 3b). Adsorbed $CO_2$ on $TiO_2\_500$-$H_2$ was thermally stable; heat treatment of this sample up to 100 °C in $N_2$ did not cause any $CO_2$ desorption from the titania surface; just some transformation of $CO_2$ bounding was noticed, such as increase the intensity of the bands at 1442 and 1353 cm$^{-1}$, which were attributed to $\delta(CH_2)$ and $\nu_s$(C-O) vibrations in the formaldehyde and formic acid, respectively.

After irradiation of $TiO_2\_500$-$H_2$ sample with UV light under flow of ethylene gas, there were observed some bands at 3668, 3632, 1725, 1622, 1552, 1469, 1442 and 1353 cm$^{-1}$, Figure 9b. The bands at 3668, 3632 cm$^{-1}$ can be assigned to OH vibrations and that one at 1725 cm$^{-1}$, to C=O vibrations in the formic acid, as compared with IR pattern spectrum (from the library HR Aldrich Solvents, CAS number: 64-18-6). Band at 1469 cm$^{-1}$ can be ascribed to formaldehyde on the basis of comparison with the pattern IR spectrum of formaldehyde (from the library HR Nicolet Sampler Library, CAS number: 50-00-0).

These measurements clearly indicated that the formaldehyde and formic acid were intermediate products in the photocatalytic decomposition of ethylene on $TiO_2$ irradiated with UV light. Similar changes in the surface chemical composition were observed on $TiO_2\_Ar$-400 sample after UV irradiation and flow of ethylene gas, Figure 9c.

Similar data were reported in the literature [11], where a mechanism of the photo-catalytic ethylene decomposition on $TiO_2$ surface was discussed, in which ethylene was transformed to formaldehyde and formic acid before complete mineralisation to $CO_2$ and $H_2O$. During this process, the quantity of hydroxyl groups on $TiO_2$ surface was decreasing. In our studies, the bands at 3600–2500 and at 1622–1625 cm$^{-1}$ were also reduced after ethylene decomposition.

## 3. Discussion

The obtained results indicated the great role of hydroxyl groups coordinated to titania surface in the photocatalytic decomposition of ethylene. $TiO_2$ samples prepared at low temperature of heat treatment, such as 400 and 450 °C in Ar, which exhibited high BET surface area and high quantity of hydroxyl groups, revealed high photocatalytic activity towards ethylene decomposition, even under UV irradiation of low intensity emitted by a fluorescence lamp. Hydroxyl groups take part in the formation of hydroxyl radicals, which are responsible for reaction with ethylene and its further degradation. The acidic surface of $TiO_2$ enhances migration of holes to the photocatalyst surface and improves separation of free charges. The presence of hole traps in $TiO_2$ formed upon hydrogen reduction could also increase adsorption of hydroxyl groups on the titania surface; however, photoluminescence spectra showed high recombination process in $TiO_2$ sample prepared at 450 °C under $H_2$, which was defected by hole traps. Interestingly, this sample showed high photocatalytic activity for ethylene decomposition under UV irradiation for higher temperature in the reactor chamber, such as 50 °C, but at conditions of 25 °C exhibited lower activity than $TiO_2$ samples prepared at low temperatures (400–450 °C) in Ar. It is assumed that at 50 °C, water molecules, which occupied active sites on titania surface $(O^{\bullet-})$, were desorbed, and at that time, reaction of peroxide anionic radicals $O_2^{\bullet-}$ with hole traps could occur to yield in $O_3^{\bullet-}$, which were responsible for ethylene oxidation. D.-R. Park et al. [4] reported that hydroxyl radicals as well as the $O_2^{\bullet-}$ and $O_3^{\bullet-}$ anionic ones played a significant role as the key active species in the complete photocatalytic oxidation of ethylene. The multilayer adsorption of water molecules on titania surface deteriorates its photocatalytic activity. However, it was demonstrated that when the photocatalytic process was conducted at higher temperature, such as 50 °C, the physisorbed water molecules could be desorbed, and then it was possible to attain the enhanced activity of $TiO_2$, as observed for $TiO_2\_A150$ sample. In general, the excess of water molecules in the gaseous stream is usually inconvenient for the photocatalytic reactions; therefore, an insignificant increase in temperature in the reaction chamber can solve the problem with competitive adsorption of

water and gaseous pollutant to the active sites of the photocatalyst. The presence of hole traps in $TiO_2$ seems to enhance its photocatalytic activity towards decomposition of VOCs (Volatitle Organic Compounds). In our previous paper [14], we have reported on enhanced photocatalytic activity of $TiO_2\_450$-$H_2$ sample towards acetaldehyde decomposition, which was caused by the presence of hole traps in $TiO_2$ that readily reacted with acetaldehyde molecules through the formation of reactive carbonyl radicals ($CH_3CO^{\bullet}$) and took place in the further stages of acetaldehyde decomposition. In case of ethylene species, their reactivity with titania surface is very weak but increases after formation of reactive radicals upon UV irradiation. Superhydrophilic properties of $TiO_2$ cause an increase in water molecules adsorption on its surface after UV irradiation. Formed thin film of water significantly limits adsorption of oxygen on the titania surface, and in this way, separation of free charges can be suppressed. Therefore, in case of ethylene decomposition, increased temperature in the reaction chamber can significantly improve the yield of the photocatalytic reactions. Application of titania samples with hole defects such as $TiO_2\_450$-$H_2$ at the increased temperature of reactor can be effective for decomposition of both VOCs, acetaldehyde and ethylene. The presence of oxygen vacancies in $TiO_2$ appeared to be detrimental for its photocatalytic activity towards decomposition of ethylene gas. It was proved that in case of the reduced $TiO_2$ prepared under hydrogenation at 500 and 600 °C, $CO_2$ adsorbed on the oxygen vacancy sites and occupied active sites for ethylene species. Therefore, titania surface with preadsorbed $CO_2$ was less active towards ethylene decomposition. Adsorbed $CO_2$ at the defect sites of $TiO_2$ was thermally stable up to 100 °C; therefore, increase reaction temperature up to 50 °C did not improve its photocatalytic efficiency.

Obtained results of ethylene decomposition on the prepared $TiO_2$ samples are very satisfying by comparison with the literature data. In Table 5, some achievements of an ethylene decomposition on titania materials by the other researchers were reported.

**Table 5.** Comparison of an ethylene decomposition on the different $TiO_2$ materials.

| Ref. | Material | Reaction Conditions | Radiation | Results |
|------|----------|---------------------|-----------|---------|
| [3] | $TiO_2$ obtained by a sol-gel and calcined at 400 °C (antase/rutile = 80/20) | 100 ppm $C_2H_4$ in air, 3 °C, flow rate 100 mL/min | UV lamp—300 W (UVA and UVB) UV flux at 12 cm =2476 mW/cm$^2$ | 90% of $C_2H_4$ decomp. per 1 g of $TiO_2$ after 1 h |
| [13] | $TiO_2$ (001) facets prepared by hydrothermal method (200 °C, 24 h) and calcined at 550 °C | 500 ppm $C_2H_4$ in air, flow rate 10 mL/min | Xenon lamp, 400 W | 17% of $C_2H_4$ decomp. after 1 h |
| [26] | $TiO_2$ obtained by plasma modification of Ti foil and oxidation | 50 ppm $C_2H_4$ in air, batch reactor | Xenon lamp, 300 W, 16 mW/cm$^2$ at 280–340 mm | 75% of $C_2H_4$ decomp. after 2 h |
| [27] | $TiO_2$ coated oriented polypropylene packaging film −10% of $TiO_2$, (30 µm) | 10 ppm $C_2H_4$ in air, batch reactor | Black light—1.5 mW/cm$^2$ Fluorescent lamp <0.05 mW/cm$^2$ | 100% of $C_2H_4$ decomp. after 3 h—Bl; 75%—after 10 days—Fl |
| [7] | Nanofibres containing 10% of $TiO_2$ | 100 ppm in air, batch reactor | UVA—2.9 µW/cm$^2$ | 45% of $C_2H_4$ decomp. after 25 h |

Presented data in Table 5 indicate that high ethylene decomposition can be obtained on $TiO_2$ based material by using high power of UV light such as xenon lamp with a power of 300–400 W [3,13,26]. Our results showed that 50 ppm ethylene in air could be decomposed during 1 h of UV irradiation (UVA-66 W) with efficiency over 90% on the surface of 0.1 g of $TiO_2$ photocatalyst. When we use close distance of UV lamp to the photocatalyst surface, then without applied cooling system, temperature in the reactor increases up to 50 °C, and then we can obtain over 90% of ethylene decomposition by using UV LED lamps $8 \times 8$ W. When we compare ethylene decomposition under irradiation of the fluorescent light, by

using $TiO_2$ prepared at 400 °C in air, we can achieve almost 100% of ethylene decomposition (10 ppm) after 1 h of irradiation, whereas the other studies indicated decomposition of the ethylene (10 ppm) of 75% after 10 days [27]. It can be concluded that the obtained $TiO_2$ from the industrial titania pulp by two step preparation process—hydrothermal treatment in autoclave at 150 °C for 1 h with following heat treatment in Ar at 400 °C—is a very promising material for ethylene decomposition. The most advantageous parameters of this material are high BET surface area and the acidic surface with high quantity of OH groups.

## 4. Materials and Methods

$TiO_2$ was obtained by a two-step preparation process: hydrothermal treatment in autoclave at 150 °C under pressure of 7.4 bar and following heat treatment at 400–600 °C under flow of Ar or $H_2$. As a source of $TiO_2$, a raw titania pulp was used, which was a semiproduct from production of the titania white in Police Chemical Factory (Poland). The industrial production of titania white in Police Chemical Factory is based on the sulphuric method; therefore, the obtained raw titania pulp contained a low percentage of the sulphuric species and was mostly amorphous. XRD measurements were performed in an Empyrean difractometer of Malvern PANanalytical Ltd. company, Almelo, Netherlands, with using a copper lamp, $\lambda = 0.154439$ nm. The measurements were performed with set up parameters of Cu lamp 35 KV and 30 mA. The mean size of anatase and rutile crystallites were calculated from the Scherrer equation by using Rietveld method. The specific surface areas of titania samples were calculated applying the BET equation in the range of partial pressure of $p/p_0 = 0.05$–$0.2$, from the nitrogen adsorption isotherms measured at 77 K using QUADRASORB Si analyzer (Quantachrome, Boynton Beach, FL, USA). Before measuring, all the samples were degassed at 150 °C for 12 h under high vacuum using MasterPrep degasser by Quantachrome. FTIR/DRS spectra of $TiO_2$ samples were measured by the reflection technique in the air atmosphere using FTIR spectrophotometer (FT/IR 4200, Jasco International Co., Ltd., Tokyo, Japan). Spectra were recorded with the resolution of 4 cm$^{-1}$. For the in situ FTIR measurements, spectrometer Nicolet iS50 was used with Praying Mantis DRIFTS accessory. UV-Vis/DR spectra were recorded using UV-Vis diffuse reflectance spectrophotometer (V-650, Jasco International Co., Ltd.,Tokyo, Japan). $BaSO_4$ was used as a reference. The amount of sulphur in $TiO_2$ was measured in energy dispersive X-ray fluorescence (EDXRF) spectrometer (Epsilon3, Malvern PANanalytical Ltd. company, Almelo, Netherlands), using internal pattern. Thermogravimetric analyses were carried out under the nitrogen flow (99.999% pure, 30 mL/min) system consisting of the thermobalance (TG, Netzsch STA 449 C, Selb, Germany). Applied temperature program was as follows: heating to 120 °C with 30-min isothermal step, then heating to 500 °C with 15-min isothermal step. Heating rate of 20 K/min was applied. The sample weight for analyses was approximately 10 mg. Mass loss that occurred at 120 °C was then presented as physisorbed hydroxyl groups and mass loss at 500 °C as chemisorbed hydroxyl groups. Photoluminescence spectra were recorded in a fluorescence spectrometer Hitachi F-2500 using low temperature sample compartment accessory. The measurements were performed at the temperature of liquid nitrogen, at excitation wavelength of 290 nm. The emission spectra were recorded in the range of 330–700 nm. The morphology of the samples was analysed by FE-SEM in SEM Hitachi SU8020 with field cold emission.

EPR spectra were recorded at the temperature of 77 K in a JEOL JES-X310 (Japan) in the quartz glass tubes under an inert gas atmosphere. Sample weights used for EPR measurements were equal, 0.2265 and 0.0665 g for $TiO_2$ reduced at 500 and 600 °C under $H_2$, respectively. The presented signals in EPR spectra were not recalculated according to the sample mass. The X-ray photoelectron spectroscopy measurements were performed in a multipurpose (XPS, LEED, UPS, AES) UHV system (PREVAC, Rogów, Poland). The spectrometer was calibrated by using Ag 3d5/2 transition. The XPS measurements were performed under vacuum at the range of 10–9 mbar after thorough degassing of sample. The X-ray photoelectron spectroscopy was performed using magnesium tube MgKα ($h\nu = 1253.7$ eV) radiation. The measurements were performed for binding energies cor-

responding to Ti 2p and O 1s regions. Results were elaborated with using the CasaXPS version 2.3.16 Dev 39. The amount of sulphur in $TiO_2$ was measured in energy dispersive X-ray fluorescence (EDXRF) spectrometer (Epsilon3, PANalytical), using internal pattern. The measurements of zeta potential at the different pH of solution and point of zero charge ($pH_{pzc}$) of the photocatalysts surface were performed in Multi-Purpose Titrator MPT-2 and degasser attached to the Zetasizer Nano-ZS. The photocatalytic decomposition of ethylene was carried out in the quartz photoreactor, which was placed in the incubator with controlled temperature. The tested sample was coated on the glass plates, which were put inside the quartz tube. The model ethylene gas of concentration 10 or 50 ppm was supplied to the photoreactor from the bottle. The photocatalytic tests in this photoreactor were carried out at the temperature of 25 °C. The scheme of the installation set up was published elsewhere [28]. Another photocatalytic test was performed in the second photoreactor, which was similar to the previous one, but applied by two sets of UV lamps (Semilac UV LED, 36 W), mounted above and beneath the bottom of the quartz tube. The scheme of the second photoreactor was illustrated in Figure 10. The difference between these two photocatalytic systems was that the second one was not equipped in any thermostatic chamber. The measured temperature inside the quartz tube during UV irradiation reached 50 °C. The emission spectrum of the UV LED lamps was measured by the USB4000 Fiber Optic Spectrometer and introduced in Figure 11.

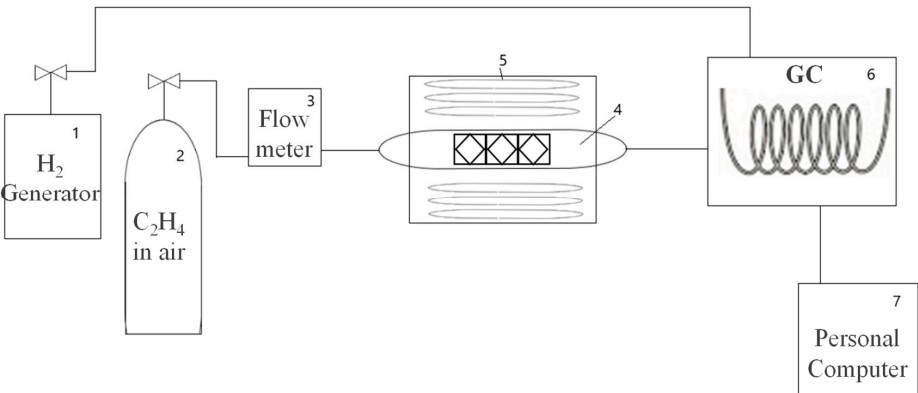

**Figure 10.** The scheme of the second photocatalytic system: 1—hydrogen generator, 2—model ethylene gas in the synthetic air (50 ppm), 3—flow meter, 4—quartz tube, 5—Semilac UV LED lamps, 6—gas chromatograph, 7—personal computer.

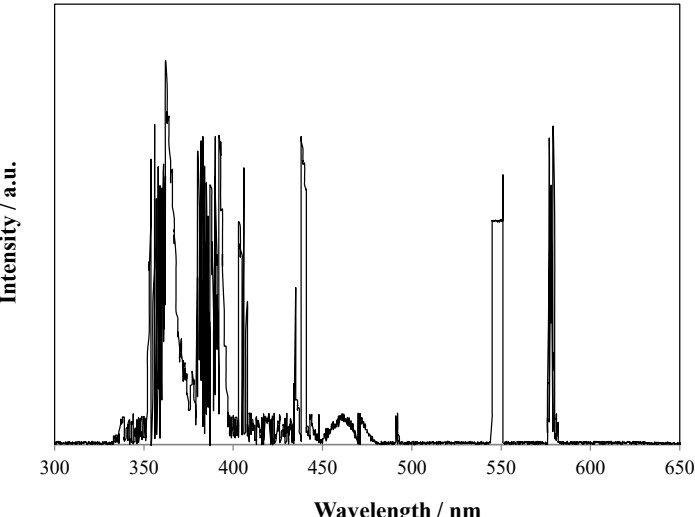

**Figure 11.** The emission spectrum of UV LED lamps.

## 5. Conclusions

Photocatalytic decomposition of ethylene is effective for $TiO_2$ having high quantity of OH groups. However, high quantity of physisorbed water molecules is disadvantageous for the photocatalytic reactions of ethylene oxidation because the presence of aqueous film on $TiO_2$ surface inhibits adsorption of oxygen and accelerates recombination of free carriers. The oxidation process of ethylene is going on mostly by the hydroxyl radicals' reaction; for total decomposition of one ethylene molecule, 12 molecules of OH radicals are used. Utilisation of OH groups on $TiO_2$ surface upon ethylene decomposition under UV irradiation was confirmed by in situ FTIR measurements. Performed studies indicated that the presence of oxygen vacancies in $TiO_2$ deteriorated its activity towards ethylene decomposition because $CO_2$ molecules were adsorbed at the sites of oxygen vacancies and occupied the active sites for ethylene species. The acidic surface of $TiO_2$ increases adsorption of OH groups and enhances activity of $TiO_2$ towards ethylene decomposition. The optimal amount of OH groups on $TiO_2$ surface retards recombination of free radicals. The excess of adsorbed water molecules can be desorbed during reaction process carrying out at higher temperature. It was proved that prepared $TiO_2$ with hole traps defects can be very active for ethylene decomposition under UV irradiation at 50 °C because at this temperature, desorption of water molecules took place, and some of $O^{\bullet-}$ species can be active for reaction with ethylene molecules or other oxygen species involved in ethylene oxidation.

**Supplementary Materials:** The following supporting information can be downloaded at: https://www.mdpi.com/article/10.3390/catal12040386/s1, Figure S1: SEM images of $TiO_2$ prepared in Ar at: (A) 400 °C, (B) 500 °C, (C) 600 °C and at $H_2$: (D) 450 °C, (E) 500 °C, (F) 600 °C, magnification ×200 k.

**Author Contributions:** P.R.: investigation, data curation, writing-original draft preparation, visualization; B.T.: conceptualization, methodology, writing-review and editing, project administration, funding acquisition; A.S.: investigation, data curation; P.F.: investigation, data curation; J.S.-N.: investigation, data curation; R.J.W.: investigation, data curation, formal analysis; H.N.: investigation, data curation; M.T.: resources, formal analysis. All authors have read and agreed to the published version of the manuscript.

**Funding:** National Science Centre, Poland, grant nr 2020/39/B/ST8/01514.

**Conflicts of Interest:** The authors declare no conflict of interest.

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
