# Peer review of "Role of the Hydroxyl Groups Coordinated toTiO2 Surface on the Photocatalytic Decomposition of Ethylene at Different Ambient Conditions"

_catalysts, doi:10.3390/catal12040386_

Round 1

Reviewer 1 Report

This manuscript reports a series of TiO2 photocatalysts for the decomposition of ethylene. The catalysts were prepared by using as starting material a previously published titania pulp which was treated at 150 °C in autoclave. Herein, TiO2 has been heated under H2 or Ar atmosphere at different temperature values to afford a series of materials which were characterized by several techniques.

I suggest a major revision of the manuscript after considering the following issues.

In order to evaluate the amount of both surface hydroxyl groups and the amount of water molecules a thermogravimetric analysis of the materials, under inert atmosphere, could be performed.

The authors claim that IR spectra of the spent catalysts (not reported) do not show any difference by comparison with those of the as-synthesized materials. In order to provide a more detailed evaluation of the surface composition of the spent catalysts an XPS study should be carried out.

In materials and methods section the sentence “Applied analytical methods such as…were widely reported in our recent paper published in Catalysts” should be replaced by the description of the methods since the characterization techniques deal with a different study.

Conclusions should be rewritten by giving more incisive closing remarks with some perspectives and comparisons on the catalytic activity of the materials with literature data.

The sentence “in our previous paper published in Catalyst” (repeated twice at pp 3, 12) is not politically correct especially considering that the manuscript should be evaluated without regard to previous publication journal. I suggest to remove these sentences and to put just the bibliographical note.

Author Response

  1. TG analyses have been performed, as suggested by the Reviewer, the results from the obtained data were added to the text.
  2. In order to analyse the composition of TiO2 surface after photocatalytic irradiation, some additional measurements were performed such as in situ FTIR with using higher concentration of ethylene gas – the changes in the chemical surface of TiO2 were observed after UV irradiation – the obtained data were added to the text and were carefully discussed

Ethylene was poorly adsorbed on the titania surface under dark conditions, the interaction of ethylene molecules with TiO2 was observed after UV irradiation. However, the obtained intermediate products were easily desorbed when the irradiation was stopped. For that reason, it was not possible to detect the reaction products outside of the reactor system, so FTIR and XPS would not show any changes in the chemical composition of titania surface before and after reaction. Therefore we performed measurements of FTIR in situ.

  1. Applied analytical methods were described, as recommended by the Reviewer.
  2. Conclusions were completely rewritten, the comparison with the other data reported in the literature was performed.
  3. The sentence “in our previous paper published in Catalyst” was delated, the bibliographical note was retained instead of this.

Reviewer 2 Report

In this paper, the authors reported the hydrothermal synthesis of anatase TiO2. The prepared samples were subjected to ethylene decomposition under visible and ultraviolet irradiation at 25 ℃ and 50 ℃. The results show that TiO2 with a large number of terminal OH groups has high photocatalytic activity, and the average grain size of anatase is about 16 nm. The presence of physically adsorbed water molecules and hole traps in defective TiO2 enhances the recombination process of free charge. This paper is well written, but needs the following revisions before publication:

  1. The abstract is too cumbersome, and the author needs to reduce the relevant content
  2. What are the advantages of this job over other jobs? The author is advised to make a table for comparison.
  3. Relevant mature things need to be deleted by the author and should not be reflected separately in the text. As shown in Figure 7 and figure 8.
  4. In order to better describe the sample, the author needs to provide some relevant tests, such as scanning electron microscope and XRD.
  5. There are too few references in this paper, and the author needs to supplement a large number of relevant references.
  6. About “High efficiency photocatalyst”, some relevant literature authors need to mention, such as: RSC Adv., 2017, 7, 25314-25324; Advanced Powder Technology, 2022, 33(3), 103481; Colloids Surf A Physicochem Eng Asp., 2022, 633(2), 127918; Appl. Catal. A-Gen., 2016, 524, 163-172; Bull. Korean Chem. Soc., 2013, 34(10), 3039-3045; RSC Adv., 2018, 8, 42233-42245.

Author Response

  1. Abstract was corrected, as recommended by the Reviewer.
  2. Comparison of the obtained data with the others, which have been already published was performed.
  3. Text referred to the Fig. 7 and Fig. 8 was rewritten.
  4. The figure with XRD patterns as well as some of the SEM images were added to the text.
  5. The number of references was extended.
  6. I am sorry, but I will not cite the mentioned by the Reviewer papers, because they are out of scope of this manuscript, instead of this I cited the other ones.

Round 2

Reviewer 1 Report

In my opinion the present manuscript has been well reviewed according to the suggestion and comments and all the critical issues have been addressed.